# Roles of Bromodomain Extra Terminal Proteins in Metabolic Signaling and Diseases

**DOI:** 10.3390/ph15081032

**Published:** 2022-08-22

**Authors:** Dayu Wu, Qiong Duan

**Affiliations:** 1Cardiovascular Division, The First Affiliated Hospital of Nanchang University, Nanchang 330006, China; 2Jiangxi Hypertension Research Institute, Nanchang 330006, China

**Keywords:** BET bromodomain, metabolism, BET inhibitors, diabetes, adipogenesis, metaflammation

## Abstract

BET proteins, which recognize and bind to acetylated histones, play a key role in transcriptional regulation. The development of chemical BET inhibitors in 2010 greatly facilitated the study of these proteins. BETs play crucial roles in cancer, inflammation, heart failure, and fibrosis. In particular, BETs may be involved in regulating metabolic processes, such as adipogenesis and metaflammation, which are under tight transcriptional regulation. In addition, acetyl-CoA links energy metabolism with epigenetic modification through lysine acetylation, which creates docking sites for BET. Given this, it is possible that the ambient energy status may dictate metabolic gene transcription via a BET-dependent mechanism. Indeed, recent studies have reported that various BET proteins are involved in both metabolic signaling regulation and disease. Here, we discuss some of the most recent information on BET proteins and their regulation of the metabolism in both cellular and animal models. Further, we summarize data from some randomized clinical trials evaluating BET inhibitors for the treatment of metabolic diseases.

## 1. Introduction

Epigenetic modulation involves the addition (writing) or removal (erasing) of histone modifications across the chromatin to facilitate changes in the heterochromatin allowing its transition into the open, activated euchromatin state necessary for transcription [1,2]; Lysine acetylation has emerged as an important and widespread regulatory posttranslational modification in several key proteins, including, the modification of various lysine sidechains within the unstructured amino-terminal tail of various histone proteins [1,2,3]. This process is closely associated with transcriptional regulation and is a key target in the evaluation of epigenetic diseases [3]. The bromodomain extra terminal (BET)- family of epigenetic reader proteins, including BRD2, BRD3, BRD4, and BRDT (hereafter referred to as BETs), regulate gene transcription via their recognition and binding of acetylated histones [4,5]. Given this critical role, BETs were treated as both a plausible and important therapeutic target, and investigations of these proteins facilitated the development of two sets of BET inhibitors in 2010 [6,7]. The development of these chemical BET inhibitors greatly facilitated and accelerated the study of BET family proteins, with several investigations revealing their close association with metabolic signaling [8,9,10] and diseases [11,12]. Therefore, we designed this review to evaluate the pathophysiological roles of these proteins in both normal metabolic processes and their disorders.

## 2. Overview of BET Bromodomain

Bromodomains, which are comprised of ~110-amino-acid modules, consist of a four-helix bundle (helices a Z, a A, a B, and a C) with a left-handed twist, and a long intervening loop between helices a Z and a A (termed the ZA loop) [13]. These bromodomain modules form a deep hydrophobic pocket that then recognizes and interacts with acetylated histone lysines, facilitating their key role in transcriptional regulation [13]. The human proteome encodes 61 BRD modules across 42 diverse proteins including several BET proteins, such as BRD2, BRD3, BRD4, and BRDT, which form a subfamily featuring two tandem bromodomains (BD1 and BD2) [3]. Both BD1 and BD2 are essential for BET activity and facilitate their interactions with acetylated chromatin [14], with each domain making a distinct functional contribution to gene transcription [15]. It was recently reported that BD1 primarily regulates steady-state gene expression, whereas the rapid increases in transcription induced by inflammatory stimuli require the activity of both the BD1 and BD2 across all of the BET proteins [15]. BRD4 is the most well-studied of these proteins and acts as the prototype for this subfamily.

The first study to report a clear connection between BETs and transcriptional regulation via their recognition of and interaction with acetylated histones was published in 2003 [14]. Further, numerous studies confirmed that BETs play a key role in transcriptional regulation. In 2005, two independent groups reported that BRD4 recruits positive transcription elongation factor b (P-TEFb) to the promoter of target genes [4,16]. P-TEFb is a heterodimer composed of cyclinT1, T2, and cyclin-dependent kinase 9 [4,16]. Upon recruitment to the promoter by BRD4, P-TEFb phosphorylates RNA polymerase II (RNAPII) inducing transcription in vivo [4,16]. In addition to recruiting P-TEFb, BET protein BRD4 also acts as an atypical kinase binding the carboxyl-terminal domain of RNAPII directly phosphorylating its serine 2 [17] and has also been shown to recognize non-histone acetylation [18]. Huang et al. reported that BRD4 binds to acetylated lysine-310 within the RelA subunit of nuclear factor (NF)-κB and enhances transcriptional activation of NF-κB and a subset of NF-κB-responsive inflammatory genes [18]. In 2013, Young et al. proposed the concept of the super enhancer, which differs from typical enhancers in size, transcription factor density and content, ability to activate transcription, and sensitivity to perturbation [19,20]. This group also identified BRD4 as a critical supporting partner for the establishment of several super enhancers known to drive the expression of various key genes associated with multiple myeloma [19]. Subsequently, other studies also reported that BRD4 is a key co-transcription factor in super enhancer-mediated gene expression across multiple biological processes [21,22]. BRD4 not only functions as a negative cotranscription factor but also a target that is positively phosphorylated or dephosphorylated by kinase or phosphatase, respectively. Wu et al. reported that casein kinase II-mediated phosphorylation of a conserved acidic region in BRD4 modulates its association with P53, modulating the expression of various P53 target genes [23]. In addition, Shu et al. found that hyper-phosphorylation of BRD4 is closely associated with resistance to BET inhibitors in triple-negative breast cancer [24]. In addition, BRD4 phosphorylation has also been shown to be involved in various cognitive activities, such as memory formation and extinction [25] (Figure 1).

These were all critical steps in establishing BET research, but these evaluations received a significant boost in 2010 when two independent groups reported structurally similar BET inhibitors in the same issue of Nature [6,7]. These studies demonstrated that BETs play a key role in controlling the expression of onco- and inflammatory genes with each of these inhibitors facilitating the functional evaluation of these BET bromodomains [6,7]. Since then, the BET bromodomain has been shown to be involved in multiple pathophysiological processes, including various cancers [26], inflammation [15,22,27], fibrosis [28], heart failure [28,29,30], and metabolic disorders [27,31]. These early BET inhibitors, including JQ1, I-BET, target both BDs equally and have similar biological effects, although they display distinct bioactivities [6,7]. RVX-208 is another important early BET inhibitor that has been shown to selectively target BD2 [32]. Recently, BD1 or BD2 selective inhibitors with better efficacy and tolerability have been developed [15]. The major BET inhibitors are listed in Table 1.

## 3. BET Bromodomain and Metabolic Signaling

Cells growing under sufficient energy conditions produce enough acetyl-CoA to support adenosine triphosphate (ATP) synthesis via the tricarboxylic acid (TCA) cycle and the cellular respiration, or it functions as a substrate to support histone acetylation. Given this, we can use histone lysine acetylation as a sort of “energy marker”.

Helping to inform the genome that energy is available for growth. Histone acetylation and its recognition by BET bromodomains are tightly coupled with energy metabolism. Therefore, the association between energy sensing signaling transducer and BET bromodomain are discussed initially.

### 3.1. Adenosine Monophosphate-Activated Protein Kinase (AMPK) and Autophagy

AMPK is a master regulator of cellular energetics. AMPK is activated in response to energy stress by sensing increases in AMP: ATP and ADP: ATP ratios. This kinase acts to restore energy balance by inhibiting anabolic processes that consume ATP, while promoting catabolic processes that generate ATP [81]. A recent study found that AMPK maintains the epigenome of mixed-lineage leukemia (MLL)-rearranged acute myeloid leukemia (AML) by linking acetyl-CoA homeostasis with BET recruitment to the chromatin. This study also showed that AMPK deletion reduces acetyl-CoA and histone acetylation, displacing BETs from the chromatin in leukemia-initiating cells, and reported that treating these cells with AMPK and BET inhibitors synergistically suppressed AML [82].

Macro autophagy/autophagy is an intracellular recycling system that delivers cytoplasmic organelles and materials to lysosomes for degradation [83]. This process is regulated by several autophagy-related (ATG) genes [84] and tightly controlled by stress-responsive signaling pathways [85]. This means, that AMPK, which acts as the primary sensor of cellular energy status, also plays a key role in promoting autophagy. In fact, a recent study has reported that BRD4 is an evolutionarily conserved autophagy repressor with BRD4 knockdown activating a series of autophagy processes including phagophore and autophagosome formation, the fusion of autophagosomes with lysosomes, and the subsequent degradation of their targets [8]. In addition, other studies have shown that nutrient deprivation disrupts the recruitment of BRD4 to the promoters of various ATG genes thus promoting autophagy. It is also worth noting that this dissociation is mediated by both AMPK and SIRT1 [8]. In summary, in the nutrient sufficient status, BRD4 suppresses autophagy that promotes growth, whereas, in the nutrient-deficient status, BRD4 facilitates cell death because the alternative source of energy, i.e., autophagy, is disrupted.

### 3.2. Yes-Associated Protein (YAP) and Transcriptional Coactivator with a PDZ-Binding Domain (TAZ)

YAP is a downstream Hippo reactive transcription factor, known for its critical role in cell growth [86]. TAZ is a YAP paralog from mammals regulated by the Hippo pathway and both YAP and TAZ are phosphorylated by LATS kinase and sequestered in the cytoplasm via their binding to 14-3-3 where they are ubiquitinated and degraded [86]. When LATS is inactive, dephosphorylated YAP/TAZ translocates to the nucleus to initiate transcription. YAP/TAZ do not contain their own DNA-binding motifs and initiate transcription by interacting with TEA domain family members 1–4 [87]. In addition to their classical effects on promoting cell proliferation, tissue regeneration, and oncogenesis, YAP/TAZ were recently shown to be involved in metabolic signaling. Accumulated data reveals that YAP/TAZ activity is strictly controlled by glucose homeostasis and lipid metabolism [88]. Glucose availability sustains YAP/TAZ activity by feeding the hexosamine biosynthesis pathway, while the reduction in glucose levels inhibits YAP/TAZ activity, mainly through the activation of AMPK [88]. Moreover, YAP/TAZ positively control various metabolic processes with White et al. revealing that YAP/TAZ promote glycolysis but suppress mitochondrial respiration [89]. In addition, YAP/TAZ have been shown to regulate adipogenesis [90] and thermogenesis [91], and a recent report suggests that YAP/TAZ physically engages with BRD4 guiding the genome-wide association of BRD4 [9]. In addition, treatment with small-molecule inhibitors of BRD4 blunts YAP/TAZ pro-tumorigenic activity in several cells and tissues making it an interesting mediator of transcriptional regulation [92]. YAP/TAZ are also key regulators of liver size and regeneration and Liu et al. found that the addition of a BET inhibitor suppresses YAP/TAZ-mediated transcription and liver regeneration [93]. However, whether other YAP/TAZ-regulated metabolic processes, such as thermogenesis, glycolysis, and mitochondrial respiration, are controlled by BETs remains unknown.

### 3.3. PGC1A

PGC-1α is a transcriptional coactivator known to control mitochondrial biogenesis and is linked to oxidative phosphorylation. PGC-1α interacts with NRF1 and 2 and stimulates mitochondrial transcription factor A, a mitochondrial matrix protein essential for the replication and transcription of mitochondrial genes. In addition, PGC-1α binds PPARα to stimulate cellular fatty acid oxidation (FAO) [94] and recent studies have shown that BRD4 coordinates with PGC-1α to control mitochondrial function and metabolism, although these outcomes and their regulation may often be in opposition under different conditions [10,95]. Padmanabhan et al. found that BRD4 interacts with GATA4 in a bromodomain-independent manner to increase the expression of PGC-1α in the heart, while BRD4 knockout suppressed PGC-1α target gene expression and mitochondrial biogenesis and function producing cardiac contractile dysfunction and lethality in various models [10]. However, in mitochondrial complex I-mutated cells, BET inhibition or BRD4 ablation actually increases oxidative phosphorylation capacity and protects against cell death [95]. This is likely mediated by the fact that BRD4 occupancy at nuclear-encoded promoters regulates the expression of mitochondrial genes and prevents PGC-1α from binding, which allows for BRD4-mediated activation of PGC-1α in normal heart tissues and PGC-1α suppression in Complex I mutated cells.

## 4. BET Bromodomain Functions in Fat Tissue Biology

### 4.1. Adipogenesis

Adipogenesis is a process where fibroblast-like progenitor cells restrict their fate to the adipogenic lineage, accumulate lipids, and differentiate into triglyceride-filled mature adipocytes. Adipocyte differentiation is controlled by the action of peroxisome proliferator-activated receptor gamma (PPARγ) and CCAAT/enhancer binding protein alpha (C/EBPα), master transcription factors known to control the gene expression program of the developing adipocyte [96]. Upon adipogenic induction, PPARγ, CEBPα, and their downstream target genes, many of which are involved in adipocyte functions such as lipid uptake and lipid synthesis, are dramatically upregulated [97]. BRD4, which is an indispensable cotranscription factor known for controlling inducible gene expression, has been shown to control proadipogenic gene expression [21]. Hu et al. reported that BETs physically associate with JMJD6 allowing for the control of its chromatin binding and proadipogenic gene transcription [98]. In addition, an analysis of BRD4 chromatin occupancy by Brown et al. revealed that the induction of adipogenesis in 3T3L1 fibroblasts provokes dynamic redistribution of BRD4 to de novo super-enhancers proximal to genes controlling adipocyte differentiation [21]. Disruption of BRD4 by siRNAs or chemical inhibitors suppresses adipogenesis, supporting a key role for BETs in adipocyte differentiation [21]. Our in vivo evaluations revealed that heterozygous BRD4 knockout in adipose tissues results in retarded body weight gain and early death (4–5 weeks) in an animal model, suggesting an indispensable role for BRD4 in fat biology [99]. This hypothesis was then further supported by the fact that treatment with JQ1 decreases both body weight and fat content [99]. In addition, Lee et al. found that BRD4 is required for the development of brown adipose tissues [100]. Although BRD2 and BRD4 belong to the BET bromodomain subfamily, it seems that they act in opposition when controlling adipogenesis. BRD2 knockdown promotes, while its overexpression suppresses, adipogenesis in 3T3L1 preadipocytes [101,102].

### 4.2. Lipolysis

Lipolysis is the process through which TAGs are hydrolyzed to release fatty acids (FA) for use by organs, such as the liver and skeletal muscle, when faced with carbon scarcity. Lipolysis requires at least three different enzymes: ATGL catalyzes the initial step of lipolysis, converting TGs to diacylglycerols (DGs); hormone-sensitive lipase (HSL) is primarily responsible for the hydrolysis of DGs to monoacylglycerols (MGs) and MG lipase hydrolyzes MGs [103]. These enzymes are regulated at both the transcriptional and post-transcriptional levels and while BETs are known to control inducible gene transcription, their role in controlling lipolysis gene transcription remains less defined. A recent study reported that overexpression of BRD2 promotes lipolysis in mice and 3T3L1 via ERK/HSL pathway activation and perilipin 1 degradation. In addition, myeloid lineage-specific BRD4 knockout promotes lipolysis in adipose tissue and leads to reduced obesity in mice [104]. Thus, while there is some indirect evidence supporting the inclusion of BETs in the regulation of lipolysis more studies are required to confirm and evaluate these roles.

### 4.3. Thermogenesis

Brown adipocytes harbor small, multilocular lipid droplets and an abundance of mitochondria, which produce heat through non-shivering thermogenesis. Heat production by BAT is governed by uncoupling protein-1 (UCP1), which resides in the inner mitochondrial membrane of brown adipocytes and functions as a long-chain fatty acid/H+ symporter to catalyze mitochondrial proton leak and thereby uncouple electron transport from ATP synthesis, with UCP1 strongly induced by cold stress [105,106]. We found that core body temperature was comparable between JQ1-treated and control mice under room temperature conditions. However, when the mice were exposed to 4 °C for 10 h, the body temperatures of the JQ1-treated mice were slightly but significantly lower than that of the control. In addition, JQ1 treatment suppressed BAT glucose uptake as determined by PET-CT scanning [99]. These data suggest that BETs may play an important role in thermogenesis, and this hypothesis was then supported by another study, which described a reduction in the thermogenic response of brown adipocytes following the addition of HDAC11. Additional evaluations revealed that this response was BRD2-dependent and an in vitro study went on to confirm that BRD2 knockdown completely blocks HDAC11 mediated suppression of Ucp1 and PGC1α mRNA, although it did not explore whether BRD2 influenced body temperature [107]. Taken together these results suggest there may be some BET-mediated regulation of thermogenesis, but more studies are needed to evaluate this role in greater detail.

### 4.4. Obesity

Obesity is characterized by increased fat accumulation, often resulting from excessive TAG storage in white adipose tissues following excessive TAG hydrolysis. Obesity leads to increased chronic low-grade inflammation, which impairs insulin signaling, contributing to the development of type 2 diabetes mellitus and cardiovascular diseases [108]. Hu et al. reported that deficiencies in BRD4 expression in myeloid lineage-specific cells protect mice from high-fat diet–induced obesity, inflammation, and insulin resistance in their adipose tissues [109]. In contrast to the effect of BRD4 knockout in myeloid lineage cells, genetic disruption of BRD2 (which reduces the expression of BRD2 protein) results in an extremely obese phenotype in mice with these animals reaching a weight of 60 g by 4 months of age and 90 g by 14 months, even when placed on a regular chow diet. Although disruption of BRD2 induced the development of severe obesity, knockout mice still presented with normal glycemia and glucose tolerance, and lower inflammation levels in their adipose tissues than control mice [102].

## 5. BET Bromodomains in Hepatic Biology

### 5.1. Hepatic Steatosis

Hepatic steatosis is characterized by triglyceride accumulation in hepatocytes and can progress to more severe pathologies such as nonalcoholic steatohepatitis, liver fibrosis, and cirrhosis [110]. Hepatic steatosis is an important risk factor for the development of hepatic insulin resistance and type 2 diabetes [110]. Yamada et al. reported that force-feeding fructose upregulated genes related to hepatic lipid accumulation, such as Cyp8b1, Dak, and Plin5 [111]. In addition, acetylation of histones H3 and H4, and BRD4 binding around the transcribed region of these fructose-inducible genes, were enhanced by fructose force-feeding [111]. Importantly, JQ1 treatment reduced the expression of these fructose-inducible genes, histone acetylation, and BRD4 binding around these genes [111]. Moreover, Chromatin Immunoprecipitation Sequencing (ChIP-Seq) using liver tissues from patients with non-alcoholic steatohepatitis (NASH) revealed that H3K27ac enrichment increases at gene loci associated with tumor necrosis factor α (TNFα) signaling inflammatory responses, epithelial-to mesenchymal transition, and IL2-STAT5, while BRD4 inhibition significantly reduced NASH-induced hepatocarcinogenesis [111]. Taken together, these data suggest a role for BETs in hepatic steatosis, but these relationships need further evaluation.

### 5.2. Hepatic Fibrosis

Fibrosis is characterized by the excessive deposition of extracellular matrix (ECM) in and around injured tissues in response to a wide variety of insults, such as inflammation, infection, and metabolic imbalance [112]. Fibrotic cells are characterized by the activation of the fibrogenic transcription programs in the myofibroblasts, which drive ECM production [28]. However, recent studies have shown that BRD4 expression is enhanced in liver fibrosis, and BETs inhibition has an anti-fibrotic effect in both the liver and other organs [112]. Ding et al. reported that BRD4 mediates a profibrotic response in activated hepatic stellate cells (HSCs), and inhibition of BRD4 blocks HSC activation into myofibroblasts, while JQ1 treatment attenuates CCl_4_ (carbon tetrachloride) exposure-induced hepatic fibrosis [113]. In addition, BET inhibition suppressed NASH, inflammation, and schistosomiasis-induced hepatic fibrosis [114,115,116], suggesting a broad-spectrum anti-fibrotic effect for these compounds.

### 5.3. HDL Biology

Plasma lipoproteins, which contain very low-density lipoprotein, low-density lipoprotein (LDL), and high-density lipoprotein (HDL), are macromolecular complexes used to transport hydrophobic lipids, cholesteryl esters, and triglycerides [117]. In addition, HDL mediates cholesterol efflux from atherosclerotic plaque via reverse cholesterol transport making it a protective lipoprotein. Apolipoprotein A-I (ApoA-I) is a critical component in HDL and is primarily produced by the liver via APOA1 gene expression. An increase in the synthesis of ApoA-I and HDL is believed to provide a new approach for treating atherosclerosis via their regulation of reverse cholesterol transport [44]. A recent study showed that administration of RVX-208, a BD2 selective BET inhibitor, for 12 weeks increased the expression of both ApoA-I and HDL-C, and promoted the production of large HDL particles, consistent with cholesterol mobilization [118]. These findings were confirmed in both a Phase 2b SUSTAIN and ASSURE clinical trials [119]. Mechanistically, BET bromodomain inhibition promotes the transcription of the APOA1 gene in human primary hepatocytes, thus improving cholesterol mobilization [44]. Thus, taken together, these studies proposed a novel therapeutic strategy for treating cardiovascular diseases, which might allow compensation for LDL-C lowering therapies widely used in the clinic.

### 5.4. Fatty Acid Oxidation, Gluconeogenesis, and Fasting Biology

During periods of fasting, the hepatic metabolism is programmed to initiate gluconeogenesis to maintain blood glucose levels and fatty acids oxidation to deliver fuel in the form of ketone bodies and produce hepakines to coordinate systemic energy homeostasis. These biological processes are tightly controlled at both the transcriptional and post-transcriptional levels. A recent report found that inhibition of BETs suppressed the expression of fibroblast growth factor (FGF) 15 in the ileum and decreased FGF receptor 4-related signaling in the liver, resulting in increased glucose production in the liver and hyperglycemia [31]. Mechanistically, impaired FGFR-4 signaling following BET inhibition results in increased expression of the gluconeogenesis and β-oxidation genes [31]. However, although the regulation of both gluconeogenesis and FAO gene transcription is important in carbon scarcity adaption, the influences of BETs on these genes in complex scenarios, such as extending fasting or ketogenic diet feeding, are not determined.

## 6. BET Bromodomains and Cardiovascular Diseases

### 6.1. Cardiac Metabolism and Heart Failure

Under normal conditions, the heart uses a large amount of energy when contracting but uses relatively little energy for growth. Thus, the heart primarily relies on the most energy-effective FAO for energy. However, in response to pressure overload, the heart experiences a shift away from FAO and to an increased reliance on glycolysis [120]. This change is also often coupled with transcriptional reprogramming characterized by an upregulation of the glycolytic genes and a downregulation of FAO gene expression [120]. Few studies have also revealed that BETs are likely to be involved in the transcriptional regulation of various pathological genes associated with murine transverse aortic coarctation [29]. The earliest study by Anand et al. found that BETs function as pause-release factors critical to the expression of pathological hypertrophic genes, and that inhibition of these BETs suppresses cardiomyocyte hypertrophy in vitro and pathological cardiac remodeling in vivo [30]. Subsequent studies confirmed that chemical inhibition of BETs suppressed heart failure in multiple models [28,29].

Unexpectedly, genetic deletion of BRD4 in cardiomyocytes leads to an acute deterioration in cardiac contractile function, culminating in dilated cardiomyopathy [121]. Mechanistically, BRD4 colocalizes with GATA4 at genes controlling mitochondrial bioenergy production, and BRD4 knockout results in a severe disruption of the cellular metabolism [10]. Moreover, they found that decreased BRD4 expression following heterozygous deletion results in delayed heart failure, which suggests that BRD4 may function as a critical protein scaffold in these cells via a bromodomain-independent mechanism.

### 6.2. BET Bromodomain in Metaflammation and Atherosclerosis

Metabolic inflammation, also known as metaflammation, is defined as low-grade, chronic inflammation orchestrated by metabolic cells in response to excess nutrients and energy [122]. Metaflammation contributes to the development of many metabolic diseases including type 2 diabetes mellitus, non-alcoholic fatty liver disease, and atherosclerosis [123]. Various papers have demonstrated that BETs are not only involved in canonical inflammation following lipopolysaccharide (LPS) exposure [27] but also control metaflammation. Our previous study showed that BETs coordinate with NF-κB to drive super enhancer production and inflammatory gene transcription [22]. Disruption of BETs by JQ1 attenuates endothelial inflammatory responses and, more importantly, atherogenic diet-induced-metaflammation and atherosclerotic lesions are significantly reduced in systems treated with JQ1 [22].

Despite this promising start, clinical trials using RVX-208 to treat atherosclerosis produced controversial outcomes. Tsujikawa et al. reported that, at least in patients with cardiovascular disorders, RVX-208 treatment reduced circulating levels of vascular inflammatory mediators, which may result in increased atherosclerotic plaque stabilization and decreased major adverse cardiac events in these patients [124]. However, another study by Nicholls et al. found that RVX-208 induced no significant increase in ApoA-I or HDL-C when compared to the placebo and that it did not induce any incremental regression in atherosclerosis [125]. Therefore, whether the application of BET inhibitors in the treatment of atherosclerosis produces any significant clinical benefit remains unclear.

## 7. BET Bromodomain in Diabetes

### 7.1. Pancreatic β Cells and Type 1 Diabetes

Pancreatic β-cells are located within the islets and produce insulin in response to enhanced glycemia and multiple neurohormonal factors. Type 1 diabetes (T1D) results from the progressive loss of pancreatic β cells as a result of autoimmune destruction [126]. BRD2 and BRD4 are highly expressed in pancreatic β-cells, where they normally inhibit β-cells mitosis and insulin transcription [102,127,128]. In fact in vitro evaluations have revealed that the specific inhibition of both BRD2 and BRD4 enhances insulin transcription, leading to increased insulin content [127]. In other studies, the evaluation of the natural history of T1D in humans and nonobese diabetic (NOD) mice reveals that these β-cells acquire a senescence-associated secretory phenotype (SASP) which is regulated by BET-mediated transcriptional control. In fact, the addition of BET inhibitor I-BET762 prevented diabetes in NOD mice and attenuated SASP in islet cells in vivo [128]. In addition, evaluations of BET bromodomain inhibitor I-BET151 significantly promoted the expansion of hPSC-derived pancreatic progenitor cells, which can be efficiently differentiated into functional pancreatic β-like cells (ePP-β cells) [129]. I-BET151 also irreversibly suppressed the development of type-1 diabetes in NOD mice by eliciting the regeneration of islet β-cells and inducing their proliferation [129]. Treatment with this compound also increases the expression of various genes encoding the necessary transcription factors for β-cell differentiation/function and induced pancreatic macrophages to adopt an anti-inflammatory phenotype.

### 7.2. Insulin Resistant and Type 2 Diabetes

Insulin resistance, which remains a major metabolic abnormality in the great majority of patients with Type 2 diabetes, is caused by altered functions within the insulin target cells and the accumulation of macrophages secreting proinflammatory mediators, such as IL-6, TNFα, IL-8, and MCP-1 [130]. BRD4 binds to PPARγ and increases the expression of Gdf3 in adipose tissue macrophages, resulting in increased fat accumulation and insulin resistance [109]. Myeloid lineage-specific BRD4 knockout mice fed a high-fat diet display reduced local and systemic inflammation and improved insulin sensitivity [109], while overexpression of BRD2 in white adipose tissues from wild-type mice induces insulin resistance [131]. In addition, reductions in BRD2 blocks insulin resistance even in severely obese mice [102]. These findings indicate that BRD2 and BRD4 enhance insulin resistance and are potential therapeutic targets for the clinical treatment of both insulin resistance and Type 2 diabetes.

Obesity and low-grade inflammation induce insulin resistance but a recent study showed that TNF-α inhibited insulin-stimulated glucose uptake in 3T3-L1 cells, while knockdown of BRD2 maintained their insulin sensitivity [102]. This is because BRD2 knockdown attenuates TNF-α-mediated inflammatory mRNA expression in adipocytes, indicating that BRD2 is required for TNF-α signaling and the initiation of insulin resistance in vitro [102]. Knockdown of BRD2 increased insulin-induced phosphorylation of IRS-1 and Akt, suggesting that these systems were exhibiting increased insulin sensitivity [131]. In addition, BRD2 suppresses Deptor expression, thereby activating the mTORC1 pathway, leading to feedback inhibition of insulin signaling and promoting insulin resistance [131]. The macrophage-mediated inflammatory response has also been implicated in the pathogenesis of insulin resistance. The bone marrow-derived macrophages (BMDMs) from global BRD2 reduced mice are less sensitive to LPS stimulation and have lower inflammatory cytokine production than the control BMDMs isolated from wild-type mice [27].

### 7.3. Clinical Evaluations of BET Inhibitors for the Treatment of Diabetes

Randomized clinical trials have recently been performed to determine the effect of BET inhibitors on diabetes. In the first trial, RVX-208 was used to treat unmedicated males with prediabetes and evaluated using an oral glucose tolerance test augmented with stable isotope tracers to facilitate postprandial plasma glucose levels, indices of insulin secretion and sensitivity, glucose kinetics, and lipolysis. These evaluations revealed that RVX-208 treatment produced a similar plasma glucose peak to the placebo, but with a more sustained elevation 30 min later. RVX-208 treatment also led to a reduction and delay in total and oral glucose secretion to the plasma and suppression of endogenous glucose production. The rate of glucose disappearance was also lower following RVX-208, with no effect on glucose oxidation or total glucose disposal [45]. These results suggest that RVX-208 delayed and reduced oral glucose absorption and endogenous glucose production, while maintaining plasma glucose levels via reduced peripheral glucose disposal. Thus, these effects may protect against the development of type 2 diabetes.

Another trial evaluated the impact of 3–6 months of RVX-208 treatment in terms of both lipid parameters and coronary atherosclerosis, and also evaluated the incidence of major adverse cardiovascular events (death, myocardial infarction, coronary revascularization, and hospitalization for cardiovascular causes). Patients treated with RVX-208 experienced fewer major adverse cardiovascular events than those treated with the placebo, and that this effect was more pronounced in patients with diabetes [132]. These results suggest that RVX-208 treated patients, particularly those with diabetes, experienced fewer cardiovascular events than the control group.

Finally, a recent phase III BETonMACE trial compared the effects of RVX-208 and placebo treatment on the incidence of major adverse cardiovascular events in 2425 patients with a recent diagnosis of the acute coronary syndrome (ACS) and diabetes. RVX-208 treated patients experienced a lower rate of the first hospitalization for heart failure, the total number of hospitalizations for heart failure, and combined cardiovascular death or hospitalization for heart failure [133]. These data indicate that RVX-208 treatment was associated with fewer hospitalizations in patients with Type 2 diabetes for heart failure and recent ACS. Taken together, these results indicate that BET inhibition may have additional clinical benefits in patients with diabetes.

## 8. Conclusion and Perspective

We summarized recent findings pertaining to BETs and their regulation of various metabolic processes and disorders(Figure 2), and the major conclusions are: 1. in vitro adipogenic differentiation is BET-dependent; 2. inhibition of BETs upregulates APOA1 gene expression, thereby increasing plasma HDL levels, which may improve atherosclerosis; 3. chemical inhibition of BETs and genetic deletion of BRD4 produce opposite results in the heart, suggesting BRD4 has an unknown scaffolding function; 4. randomized clinical trials of BET inhibitors in diabetes are promising.

Despite exciting progress regarding BETs and their function in the metabolism, some important issues have not been addressed. These include the fact that how BETs couple with energy metabolism for adaptive gene transcription and their involvement in the hepatic fed and fasting response remain unknown. In addition, the roles and selectivity of BD1 and BD2 in both metabolic signaling and disease progression also remain unexplored. Although there has been significant progress in this field, there are many more fundamental questions that still need to be answered. In addition, translating these findings to the clinical setting should be the focus of future studies.

## Figures and Tables

**Figure 1 pharmaceuticals-15-01032-f001:**
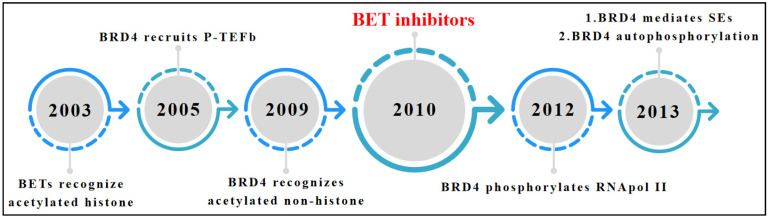
The development process of BET bromodomain proteins. P-TEFb: Positive Transcription Elongation Factor b. SEs: Super Enhancers.

**Figure 2 pharmaceuticals-15-01032-f002:**
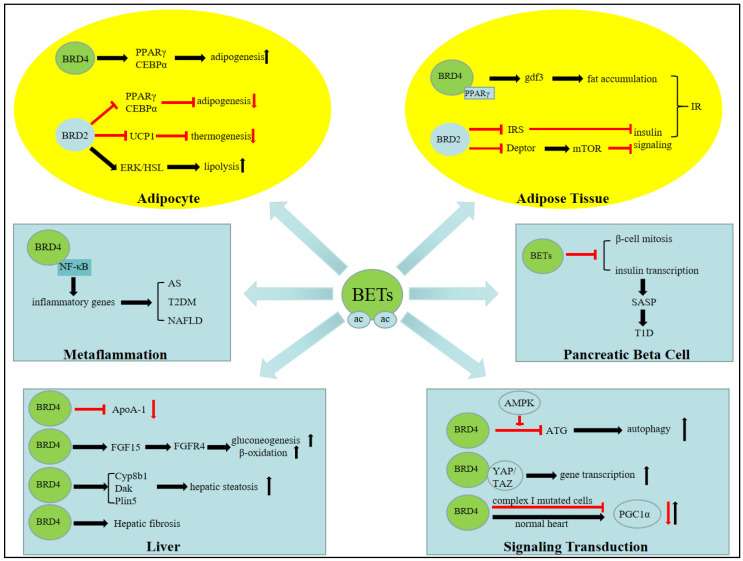
Role of BETs in metabolic processes and disorders. IR, insulin resistance; AS, atherosclerosis; T2DM, type 2 diabetes mellitus; NAFLD, nonalcoholic fatty liver disease; SASP, senescence-associated secretory phenotype; T1D, type 1 diabetes.

**Table 1 pharmaceuticals-15-01032-t001:** Summary of Drugs Targeting BET Proteins.

	Compound	Selectivity	Indication	Reference
BET inhibitors	JQ1	BD1 and BD2 from BRD2/3/4 and BRDT	Chronic obstructive pulmonary disease	[33]
			NUT midline carcinoma	[6]
			Multiple myeloma	[34]
			Acute myeloid leukemia	[35]
			Diffuse large B-cell lymphoma	[36]
			Hematologic malignancies	[37]
			Lung cancer	[38]
			Breast cancer	[39]
			Colon cancer	[40]
			Pancreatic ductal adenocarcinoma	[41]
			Colorectal cancer	[42]
			Hepatocellular cancer	[43]
	RVX-208	BD2 from BRD2,3,4	Atherosclerosis	[44]
			Diabetes	[45]
			Fabry disease	[12]
			Chronic kidney disease	[46]
	IBET-762 (GSK-525762)	BD1 and BD2 from BRD2,3,4,T	Neoplasm	[12]
			Testis carcinoma	[47]
			Midline carcinoma	[12]
	IBET-151	BD1 and BD2 from BRD2,3,4,T	MLL-fusion leukemia	[48]
			Colorectal ctumorsancer	[42]
			Gastric cancer	[49]
			Vismodegib-resistant esophageal adenocarcinoma	[50]
			Rheumatoid arthritis	[15]
			Melanoma	[51]
			Myeloma	[52]
	MK8628/OTXO15	BRD2,3,4	Lymphoma or multiple myeloma	[53]
			Acute leukemia	[54]
			NUT midline carcinoma	[12]
			Triple-negative breast cancer	
			Lung cancer	
			Castration-resistantprostate cancer	
	FT1101	BRD2,3,4,T	Acute myeloid leukemia	[12]
			Non-Hodgkin lymphoma	
	CPI-0610	BD1 from BRD2,4,T	Multiple myeloma	[12]
	ABBV-075 Mivebresib	BRD2,3,4	Relapsed/Refractory solid tumors.	[55]
			Breast cancer	[12]
			Prostate cancer	
			Non-Hodgkin lymphoma	
			Multiple myeloma	
			Relapsed/refractory acute myeloid leukemia.	[56]
	NHWD-870	BRD4	Pancreatic ductal adenocarcinoma	[41]
			Osteosarcoma	[57]
	BMS-986158	undisclosed	Advanced tumors	[12]
	PFI-1	BRD2,4	Acute leukemia	[58]
	ABBV-744	BD2	prostate cancer	[59]
			Acute myeloid leukemia	[59]
	GSK788	BD1	Acute myeloid leukemia	[15]
	GSK620	BD2	Rheumatoid arthritis	[15]
			Psoriasis	[15]
			Non–alcoholic fatty liver disease	[15]
	RO6870810/TEN-10	undisclosed	Acute myeloid leukemia and myelodysplastic syndrome	[60]
			NUT carcinoma, other solid tumors, or diffuse large B-cell lymphoma	[61]
			Multiple myeloma	[62]
	BAY 1238097	BRD4	Advanced malignancies	[63]
			Pancreatic ductal adenocarcinoma Non-small cell lung cancer	[64]
			Lymphoma	[65]
	ZEN-3694	BD1,BD2	Metastatic castration-resistant prostate cancer	[66]
	INCB054329	BRD4	Advanced malignancies	[67]
	INCB057643	BRD4	Advanced malignancies	[67]
	ODM-207	BRD2,3,4,T	Castration-resistantprostate cancer.	[68]
	AZD5153	BRD4	Malignant solid tumor and lymphoma	NCT03205176
	CC-90010	BRD2,4	Advanced solid tumors and relapsed/refractory Non-Hodgkin’s lymphoma.	[69]
			Solid tumor	[69]
BET degraders	ARV-825	BRD2,3,4,T	Burkitt’s lymphoma	[70]
			Multiple myeloma	[71]
			Secondary (s) acute myeloid leukemia	[72]
	dBET1	BRD2,3,4	Leukemia	[73]
	ARV-763	BRD4	Multiple myeloma	[71]
	ARV-771	BRD2,3,4	Castration-resistant prostate cancer	[74]
			Hepatocellular carcinoma	[75]
			Non-small cell lung cancer	[76]
			Post-myeloproliferative neoplasm secondary acute myeloid leukemia	[77]
	QCA570	BRD4	Acute leukemia	[78]
	BETd-246/BETd-260	BRD4	Triple-negative breast cancer	[79]
	MZ1	BRD4	Castration-resistant prostate cancer	[80]

The compounds are listed according to their (1) mechanism of action in blocking BETs function; (2) selectivity for BET family proteins; (3) condition or diseases for which they are being studied. BET indicates bromodomain extra terminal.

## Data Availability

No new data were created or analyzed in this study. Data sharing is not applicable to this article.

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
