# Peer review of "Roles of Bromodomain Extra Terminal Proteins in Metabolic Signaling and Diseases"

_pharmaceuticals, 2022, doi:10.3390/ph15081032_

Round 1

Reviewer 1 Report

1. In Table 1, the indications are unified with uppercase letters in pp 3~5.

2. In pp 10, beta ->β

Author Response

Thank you for your comments, and we have corrected them.

Reviewer 2 Report

This is a very thorough, well researched and well summarized review of the role of BET proteins in a variety of physiological settings. Some minor revisions pertaining to grammar are needed.

Specific Comments

- the author list appears incomplete

- Figure 1 isn’t necessary, as what is depicted was 

Author Response

Response 1: Thanks. It is a mistake; we have corrected it.

Response 2: Thank you for the comment. The purpose to use this figure is to depict the key events in BET bromodomain studies. The figure is corresponding to the text of “overview of BET bromodomain”. We think that the figure will help the readers easier to understand the progress of BET bromodomain study.

Reviewer 3 Report

1. It would be useful to have some figures for the biochemistry.  The authors go over this in some detail but a figure would help

2.  There is a language that could be corrected.  The overall meaning is clear, but it could be improved.  Would recommend an editor for this paper

Author Response

Point 1: It would be useful to have some figures for the biochemistry. The authors go over this in some detail but a figure would help

Response 1: Thank you for the suggestion. We would like to add a figure to show the chemical structure of BET bromodomain and their inhibitors, however, these information were described in detail in a lot of review articles. Thus, we think it is not necessary to include a figure in the present review.

Point 2: There is a language that could be corrected. The overall meaning is clear, but it could be improved. Would recommend an editor for this paper

Response 2: Thank you for the suggestion. We corrected some obvious mistakes in grammar. We actually had our manuscript modified by professional language editing company before submitting. If more language modification is considered to be necessary, please feel free to let us know.

Reviewer 4 Report

In this review, Wu and Duan discuss the role of BET proteins in various metabolic pathways and associated diseases.

Authors address the role and the molecular functions of BET proteins in fat tissue, liver, pancreatic and cardiovascular system thoroughly collecting and commenting on both in vitro and in vivo experiments and clinical trials results. 

The article is well written, and clear and the subject covered could be interesting for many readers.

There could be only a potential issue. In the second section of the manuscript, authors discuss the discovery of BET inhibitors as the main event for further studies on BET proteins. They also provide a table with BETi associates to disease models. This step suggests the relevance of BETi both for molecular research and for clinical practice. Nevertheless, in the subsequent sections, BETi are little or not mentioned at all. Maybe some more information about the possible association between BETi and a specific "metabolic" disease could be interesting for readers. For example, as authors have already done for diabetes in section 7.3.

Author Response

Response 1: Thank you for the suggestion. The discovery of BET inhibitors is the key step in BET bromodomain study history. We described in detail about BETi in the animal studies and clinical practices, although we did not use nomenclature of BETi. JQ1 or RVX-208, which are widely used BET inhibitors, were frequently mentioned in the whole text.